# A course-based undergraduate research experience examining neurodegeneration in *Drosophila melanogaster* teaches students to think, communicate, and perform like scientists

**Rebecca Delventhal**[1]*, **Josefa Steinhauer**[2]*

1 Department of Genetics and Development, Columbia University Medical Center, New York, New York, United States of America, 2 Department of Biology, Yeshiva College, Yeshiva University, New York, New York, United States of America

* rd2744@cumc.columbia.edu (RD); jsteinha@yu.edu (JS)

**Data Availability Statement:** The data underlying the results presented in this study are attached as Excel files in the Supporting Information.

## Abstract

As educators strive to incorporate more active learning and inquiry-driven exercises into STEM curricula, Course-based Undergraduate Research Experiences (CUREs) are becoming more common in undergraduate laboratory courses. Here we detail a CURE developed in an upper-level undergraduate genetics course at Yeshiva University, centered on the *Drosophila melanogaster* ortholog of the human neurodegeneration locus *PLA2G6/PARK14*. *Drosophila PLA2G6* mutants exhibit symptoms of neurodegeneration, such as attenuated lifespan and decreased climbing ability with age, which can be replicated by neuron-specific knockdown of *PLA2G6*. To ask whether the neurodegeneration phenotype could be caused by loss of *PLA2G6* in specific neuronal subtypes, students used *GAL4-UAS* to perform RNAi knockdown of *PLA2G6* in subsets of neurons in the *Drosophila* central nervous system and measured age-dependent climbing ability. We organized our learning objectives for the CURE into three broad goals of having students think, communicate, and perform like scientists. To assess how well students achieved these goals, we developed a detailed rubric to analyze written lab reports, administered pre- and post-course surveys, and solicited written feedback. We observed striking gains related to all three learning goals, and students reported a high degree of satisfaction. We also observed significantly improved understanding of the scientific method by students in the CURE as compared to the prior year's non-CURE genetics lab students. Thus, this CURE can serve as a template to successfully engage students in novel research, improve understanding of the scientific process, and expose students to the use of *Drosophila* as a model for human neurodegenerative disease.

## Introduction

Recent high profile reports have called for the reorganization of undergraduate STEM curricula around active learning strategies in order to improve undergraduate education, prepare students for the fast pace of discovery today, and increase the number of STEM trainees

**Funding:** Project funding provided by NIH (Eunice Kennedy Shriver National Institute for Child Health and Human Development) grant R15-HD080511-02 to J.S. (https://www.nichd.nih.gov/) The funders had no role in study design, data collection and analysis, decision to publish, or preparation of the manuscript.

**Competing interests:** The authors have declared that no competing interests exist.

entering the workforce (visionandchange.org) [1,2]. One common directive emerging from these reports is increased student participation in inquiry-based learning and research experiences. Course-based Undergraduate Research Experiences (CUREs) address this goal by delivering authentic research projects within laboratory courses, thereby serving more students from a broader range of backgrounds than apprentice-based research internships [3–7]. CUREs have been shown to increase student learning as measured by pre- and post-course skills and knowledge inventories [8–11] and to have an impact on students' conceptions of scientific research, sense of independence, and persistence in STEM majors and careers [12–15]. CUREs also confer demonstrated benefits on faculty members by advancing their research objectives, and on the broader scientific community through discovery of novel scientific results[16]. As described previously, a CURE is distinguished from other inquiry-driven exercises by inclusion of the following five elements: scientific practices, discovery, broader relevance, collaboration, and iteration [3,12].

Neurodegenerative diseases represent a growing global healthcare burden. Patients with neurodegenerative disease display a progressive loss of neurological function, leading to devastating impacts on memory and motor function, which in turn can place large socioeconomic strain on the healthcare system and caregivers. Parkinson's disease in particular affects 1% of individuals 65 years or older and is characterized by a progressive loss of coordinated movement[17]. Although most Parkinson's disease cases are sporadic, presumably deriving from a combination of genetic and environmental factors, instances of inherited familial parkinsonism have allowed for the identification of single gene mutations that can cause similar disease [18]. Elucidation of underlying cellular and molecular mechanisms in rare inherited forms may provide insight into the more common sporadic forms. To this end, numerous animal models have been generated with mutations in genes orthologous to human loci implicated in familial parkinsonism [19]. One of these is *PARK14/PLA2G6*, a disease locus for a collection of neurodegenerative disorders including early-onset dystonia-parkinsonism (collectively called *PLA2G6* Associated Neurodegeneration, PLAN) [20, 21]. A null mutant in the *Drosophila melanogaster PLA2G6* ortholog (also known as *iPLA2-β* or *CG6718*) generated in the Steinhauer lab exhibits a progressive loss of locomotor ability with age, consistent with neurodegeneration (J.S., manuscript in preparation). We and others were able to reproduce the locomotor defect with *PLA2G6* RNAi knockdown in neurons only [22, 23].

This report details a CURE developed at Yeshiva College of Yeshiva University (YU), a private four-year liberal arts college with an all-male student body. The research objective of this CURE was to investigate whether the locomotor decline observed in *PLA2G6* mutants is due to the loss of *PLA2G6* in specific neuronal subsets. To test this, students performed RNAi knockdown of *PLA2G6* in groups of neurons representing the major excitatory neurotransmitter pathways in the fly central nervous system (CNS), and in the insulin producing cells of the CNS [24–26]. This genetic manipulation used the classic *GAL4-UAS* binary gene expression system, which employs an endogenous cell-type specific promoter to direct expression of the yeast GAL4 transcription factor, which then activates expression of a *UAS*-controlled transgene, in this case RNAi targeting *PLA2G6*, in the cells of interest [27]. To monitor locomotor decline, students assessed the flies' climbing ability using a common and straightforward behavioral assay that does not require any expensive equipment and is ideal for undergraduates (adapted from [28, 29]).

The CURE was embedded in an upper-level elective Genetics course. In prior years of the course, the laboratory used *Drosophila* to conduct a semester-long three factor mapping exercise in which students mapped known mutations using recombination frequencies, an inquiry-driven project that did not involve novel research. In contrast, our CURE was centered on an open-ended, authentic research question and incorporated several novel aspects relevant

to the key elements of a CURE listed above, with a strong emphasis on discovery and broader human health relevance [3, 12]. Furthermore, we used the CURE elements to inform our learning goals for the course, which were for students to learn to (a) think like a scientist, (b) communicate like a scientist, and (c) perform like a scientist (i.e., perform science) [12]. We detail here our syllabus (Table 1) and our framework for assessing student performance within each of the learning goals according to specific learning objectives (Table 2). In order to investigate how the CURE compares to the prior syllabus, we developed and applied a rubric to written lab reports from both the CURE and the prior year's mapping project and present a comparison analysis. To assess student attitudes about the CURE, we compared self-reported student outcomes pre- and post-course using the CURE survey [13, 30] (https://www.grinnell. edu/academics/resources/ctla/assessment/cure-survey) and solicited narrative feedback. All three assessment methods indicate positive effects. Overall, we found that the CURE generated large student learning gains, fostered active learning, which is lacking within our biology major curriculum, and improved student performance in written assignments when compared

**Table 1. Course outline.**

| Week | Course Activities | |
|------|-------------------|---|
| 1 | CURE pre-course survey[a]<br>Mini-lecture/class activity: Introduction to *Drosophila* research and techniques; introduction to research project background and experimental design<br>Syllabus review: Trivia game | |
| 2 | Mini-lecture: Review experiment design<br>Lab demo: *Drosophila* husbandry | Lab tasks[b]: Receive parental fly stocks; train to identify males and females, virgins, and phenotypic markers; pass all stocks.<br>During the week: Collect virgin and male parents. |
| 3 | Lab tasks: Virgin collection and cross preparation; stock maintenance; set experimental crosses.<br>During the week: Virgin and male parent collection; pass crosses. | |
| 4 | Mini-lecture: Assays for studying neurological dysfunction in *Drosophila* | Lab tasks: Virgin collection and cross preparation; stock maintenance; set/pass crosses; learn climbing assay.<br>During the week: Pass crosses. |
| 5 | Mini-lecture: How to read scientific articles<br>Homework #1: Read assigned articles and complete worksheet. | Lab tasks: Virgin collection and cross preparation; stock maintenance; pass crosses; learn climbing assay; begin collecting F1 experimental flies.<br>During the week: Collect F1; pass crosses. |
| 6 | **Homework #1 due**<br>**Class activity: Journal Club #1** | Lab tasks: Collect F1; begin aging flies.<br>During the week: Collect F1; pass crosses. |
| 7 | Mini-lecture: Data collection and analysis | Lab tasks: Collect F1; age flies. |
| 8 | Lab tasks: Age flies; climbing assays on F1 flies. | |
| 9 | Lab tasks: Age flies; climbing assays on F1 flies. | |
| 10 | Mini-lecture: How to give a good scientific presentation<br>Homework #2: Read assigned articles and complete worksheet. | Lab tasks: Climbing assays on F1 flies. |
| 11 | **Homework #2 due**<br>**Class activity: Journal Club #2**<br>Homework: Prepare presentation of results. | |
| 12 | **Class activity: Student group presentations** | |
| 13 | **Class activity: Student group presentations**<br>CURE post-course survey | |
| 15 | **Individual lab reports due** | |

[a]Blue indicates classroom activities;

[b]green indicates laboratory activities.

Table 2. **Learning goals, objectives, and aligned activities.**

| Learning Goal | Learning Objectives | Lab Activity (numbers refer to learning objectives) | Core Elements of CURE |
|---|---|---|---|
| Think like a scientist | 1. Understand the background/context of the experiment<br>2. Understand the experimental design (controls, how the tools work, etc.)<br>3. Understand that real data is "messy"<br>4. Understand importance of replication and sample size<br>5. Understand that there are multiple explanations for a given result<br>6. Propose future experiments | -Introductory class (1,2)<br>-Semester-long experiment (3,4)<br>-Oral presentation (5,6)<br>-Final written report (5,6) | -Use of scientific practices<br>-Discovery<br>-Iteration |
| Communicate like a scientist | 1. Engage in informal discourse with peers and instructor about results and data collection<br>2. Discuss primary literature<br>3. Deliver oral presentation<br>4. Produce written report | -Semester-long experiment (1)<br>-Journal club (1,2)<br>-Oral presentation (1,3)<br>-Final written report (2,4) | -Use of scientific practices<br>-Collaboration |
| Perform like a scientist | 1. Learn research techniques<br>2. Work collaboratively<br>3. Engage in open-ended inquiry<br>4. Repeat/iterate experiments<br>5. Review primary literature<br>6. Collect and analyze data | -Semester-long experiment (1,2,3,4,6)<br>-Journal club (5)<br>-Oral presentation (2,5,6)<br>-Final written report (5,6) | -Use of scientific practices<br>-Discovery<br>-Collaboration<br>-Iteration |

to the prior inquiry-driven lab course. Thus, our course can serve as a model for development of similar CUREs to engage students in novel research, incorporate discovery-based learning into biology curricula, and involve students in the use of *Drosophila* as a model for neurodegenerative disease.

## Methods

### Course structure

**Experimental activities.** Students conducted a semester-long experiment to investigate a neuronal subtype-specific role of *PLA2G6* in age-induced locomotor decline using *Drosophila melanogaster* (Table 1). Each student pair was responsible for testing *PLA2G6* RNAi knockdown using one *GAL4* driver from the following set: *ple-GAL4* (dopaminergic), *ChAT-GAL4* (cholinergic), *tdc2-GAL4* (tyraminergic and octopaminergic), *dilp2-GAL4* (insulin-producing neurons), *VGlut-GAL4* (glutamatergic), and *elav-GAL4* (pan-neuronal). Student pairs were given the opportunity to select their *GAL4* drivers through a lottery system. At the beginning of the semester, students received their fly stocks and were trained to collect male and virgin female flies and distinguish balancer chromosome markers Curly and Stubble. Students established crosses between their *GAL4* driver line and the *UAS-RNAi-PLA2G6* line to produce F1 knockdown flies. In parallel, students established cultures of a negative control (isogenic wild-type control for *PLA2G6* mutant) that climbs normally and a positive control (*PLA2G6* null mutant generated previously in the Steinhauer lab) that exhibits age-induced locomotor decline by 20–30 days of age. Control cultures were reared in parallel with experimental crosses. All flies were kept at 26˚C to enhance *GAL4* activity. Students were instructed to pass parental cultures to fresh food at least 2x per week for 2–3 weeks to ensure a large window of F1 experimental fly eclosion. Students collected F1 males at least 2x per week to ensure that all individuals within assay fly cohorts were within a 3–4 day age range. Students were instructed to pass collected experimental flies to fresh food 2x per week to ensure health of flies during

aging. Flies were tested for climbing over a 4-week period, at both young (<1 week) and various aged time points (20–35 days).

**Instructional activities.** The semester began with an introductory lecture in which students were acquainted with the following concepts: history of *Drosophila* as a model organism, the specific genetic tools that were used in the experiment (*GAL4-UAS*, RNAi), *PLA2G6*, and neurodegenerative disease. In an iterative fashion, the instructor presented preliminary data from the Steinhauer lab and facilitated small group discussions to brainstorm follow-up experiments, which eventually led students to the neuronal subtype knockdown experiment. Students were not informed prior to the first class that they would be conducting authentic research in the lab section.

Throughout the semester, students were periodically presented with content relevant to the research through short mini-lectures during laboratory time (Table 1). Students also participated in two journal club discussions, in which they were required to read a primary research article and submit a written assignment prior to the day of the discussion (Table 1). The journal clubs were structured as "jigsaw discussions," with question handouts to guide discussion (see Supporting Information). To conclude the semester, the students presented their work in group presentations, and each student completed an individual written report in the format of a scientific research article.

**Learning goals and objectives.** Our overarching learning goals were for students to learn how to think, communicate, and perform like scientists. Each of these learning goals encompasses several learning objectives crucial to understanding and experiencing how "real" scientific research is conducted (Table 2). For example, because reviewing primary literature is an important aspect of performing science, students were required to participate in journal club discussions and use recent literature in preparation of their final written reports.

These learning goals and objectives were created to align with the core elements of a CURE as defined by Auchincloss [3] and elaborated by Brownell and Kloser [12]: use of scientific practices (thinking and communicating as a scientist, using the tools of a scientist), discovery (open-ended inquiry, illustrating the "messiness" of real data), broadly relevant or important work (to students' lives and the scientific community), collaboration (with each other, the class, the instructor, with opportunities for discourse and accountability), and iteration (understanding the need for replication and importance of sample size, distinguishing between failed experiments and negative results). In Table 2, we detail the learning goals, their associated learning objectives, and how these align with each core CURE element.

**Assessment.** In order to assess student outcomes, we employed three approaches.

**CURE survey.** We administered the CURE survey, developed by David Lopatto [13, 30] (https://www.grinnell.edu/academics/resources/ctla/assessment/cure-survey), both pre- and post-course, using Qualtrics online survey software to anonymize responses (n = 14, see Supporting Information for raw data). The pre-course survey was administered in class before any course instruction began, and the post-course survey was administered during the last class session, after all group presentations had been completed but before final reports were due. Students provided informed consent before taking the survey, and IRB oversight was provided by Albert Einstein College of Medicine. To compare scores to the national benchmark data (https://www.grinnell.edu/sites/default/files/docs/2019-07/CUREBenchmarkStatistics2015-2108.pdf), we performed a one-sample t-test, using the national benchmark as the population mean. "Positive" science attitudes were assessed by summing scores for the following five statements (total possible score of 25), which have been found previously to factor together in a principal component factor analysis (personal communication with D. Lopatto): "*Even if I forget the facts, I'll still be able to use the thinking skills I learn in science*"; "*The process of writing in science is helpful for understanding scientific ideas*"; "*I get personal satisfaction when I solve a*

*scientific problem by figuring it out myself"; "I can do well in science courses"; "Explaining science ideas to others has helped me understand the ideas better."* Similarly, "negative" attitudes were assessed by summing the scores for the following six statements (total possible score of 30): *"I wish science instructors would just tell us what we need to know so we can learn it"; "Creativity does not play a role in science"; "Science is not connected to non-science fields such as history, literature, economics, or art"; "Science is essentially an accumulation of facts, rules, and formulas"; "There is too much emphasis in science classes on figuring things out for yourself"; "If an experiment shows that something doesn't work, the experiment was a failure"* [31, 32].

**Lab report scoring.** We developed a rubric for evaluating student learning outcomes from final written lab reports, adapted from [33–37] (see Supporting Information for rubric and raw data). The rubric included subcategories that we coded according to our learning goals for students: think, communicate, and perform like scientists. We also quantified performance on understanding of biology concepts and principles of the scientific method, by coding statements throughout the written lab reports that related to either the specific experimental details of this project or general principles of experimental design. To develop a rubric with high inter-rater reliability, the two graders (R.D. and J.S.) completed two rounds of test scoring in which both authors scored six reports (three from 2017 and three from 2018), after which scores were discussed and the rubric revised. Both authors then independently applied the finalized rubric to all the written reports from both the fall 2018 semester (n = 14) and the prior year's inquiry-based mapping laboratory (n = 24). Inter-rater reliability was measured with a Cohen's weighted kappa value of 0.71 for the overall scores. Individual graders' scores were averaged to derive final scores. To compare scores between the 2017 and 2018 reports, we performed a Kruskal-Wallis analysis with Dunn's correction for multiple comparisons.

**Narrative feedback.** We used an online survey to solicit anonymous, narrative feedback from students on their experiences and reactions to the course, after the conclusion of the course but before grades were distributed.

***Drosophila* stocks and husbandr.** *ple-GAL4 (BDSC8848, RRID:BDSC_8848), VGlut$^{OK371}$-GAL4 (BDSC26160, RRID:BDSC_26160), ChAT$^{7.4}$-GAL4 (BDSC56500, RRID:BDSC_56500), tdc2-GAL4 (BDSC9313, RRID:BDSC_9313), dilp2-GAL4 (BDSC37516, RRID:BDSC_37516),* and *HMS01544 (BDSC36129, RRID:BDSC_36129)* were from Bloomington *Drosophila* Stock Center. *elav-GAL4* was a gift from J. Treisman. *PLA2G6* null mutants and isogenic controls were generated in the Steinhauer lab (J.S., in preparation).

*Drosophila* were cultured on standard media containing 3.83% molasses, 1.58% yeast, and 3.83% corn meal, supplemented with 0.11% methyl paraben and 0.38% propionic acid as mold inhibitors.

**Climbing assay.** Climbing tests were adapted from [28, 29, 38] and performed at room temperature (see Supporting Information for protocol). Groups of 6–12 male flies were tapped to the bottom of a fresh food vial and given 20 seconds to climb into a new empty vial placed on top (6 cm). Each group of flies was given five climbing trials per assay. Each fly in the group was assigned one point for every success at climbing out of the bottom vial, and the total number of points for the group was divided by the number of flies in the group to yield the climbing index. Students were instructed to assay at least four groups per condition, and results were averaged.

## Results

### Experimental results

Fourteen students completed the course, and they worked in self-selected pairs for the entirety of the semester. Students collected and assayed at least four groups of 6–12 flies each for each

genotype (isogenic control, *PLA2G6* mutant, or *PLA2G6* knockdown) at young age (<10 days old) and at 22–28 days old. Because the experiment required students to attend to their flies between planned class sessions, students were given the autonomy to arrange their experiments according to their schedules. For the young flies, five out of seven student pairs tested at least two climbing cohorts for each genotype, while one pair failed to test the knockdown flies and one pair tested only one climbing cohort for each genotype. For the aged flies, six out of seven student pairs tested at least two climbing cohorts for each genotype, while one pair tested only one *PLA2G6* mutant cohort. Although not explicitly instructed to do so at the beginning of the course, four out of seven student pairs elected to test flies aged beyond 30 days. Because by this time point many of the *PLA2G6* mutant flies had died due to shortened lifespan [22, 23, 39], only negative control and knockdown flies could be assayed reliably. Two examples of student-generated data are shown in Fig 1. Student results for the null mutant and isogenic control were consistent with prior results from the Steinhauer lab, with climbing activity in the mutant initially normal but reduced by 20 days of age (Fig 1, J.S. in prep). In flies aged between 20–29 days, only pan-neuronal and cholinergic knockdown displayed attenuated climbing, but neither reached statistical significance (Fig 1 and not shown). In flies aged >30 days, cholinergic knockdown resulted in significantly reduced climbing activity, in contrast to knockdown in dopaminergic or tyraminergic and octopaminergic neurons (Fig 1 and not shown). Pan-neuronal knockdown flies aged >30 days appeared to show reduced climbing as well, consistent with prior data [22, 23] (J.S. in prep), but only one cohort was assayed, precluding statistical analysis for this genotype. Together, the students' results suggest that *PLA2G6* depletion in cholinergic neurons may lead to age-induced locomotor decline.

## CURE survey results

**Students report high levels of prior experience with non-active, traditional course elements.** In the pre-course CURE survey, students were asked to rate their level of prior experience with different types of course elements or activities. These ranged from traditional approaches, such as "take tests in class" and "work individually," to more interactive learning approaches, such as "work as a whole class" and "a lab or project where no one knows the outcome." We found that our students rated a very high level of experience in the most traditional, inactive methods of instruction (Fig 2A): take tests in class (4.8, all scores have a maximum of 5), read a textbook (4.7), listen to lectures (4.3), and perform a cookbook lab with an expected outcome (3.7). When compared to national benchmark data gathered from students enrolled in CUREs between 2015–2018 (https://www.grinnell.edu/sites/default/files/docs/2019-07/CUREBenchmarkStatistics2015-2108.pdf), our students rated a significantly higher level of experience in taking tests and reading textbooks. This suggests that our students began the course with a more traditional academic background, with less experience in active forms of learning compared to students at other institutions. Consistent with this, in several course elements representing more active forms of learning (Fig 2B), our students reported a low level of prior experience, including in projects of student design (1.6) and critiquing the work of other students (1.8), despite having moderate to high experience with problem sets (3.5) and discussing reading materials in class (4.4). Because most of the students in this course were upper-level biology majors (S1 Fig), these results suggest that the YU biology curriculum was lacking active learning components prior to the implementation of this CURE.

**Students report large benefits and gained experience in learning goals (scientific thinking, communication, and skills).** To understand whether we achieved our learning goals of having students learn to think, communicate, and perform like scientists, we examined students' survey responses of experience gained in course elements (pre- and post-course data)

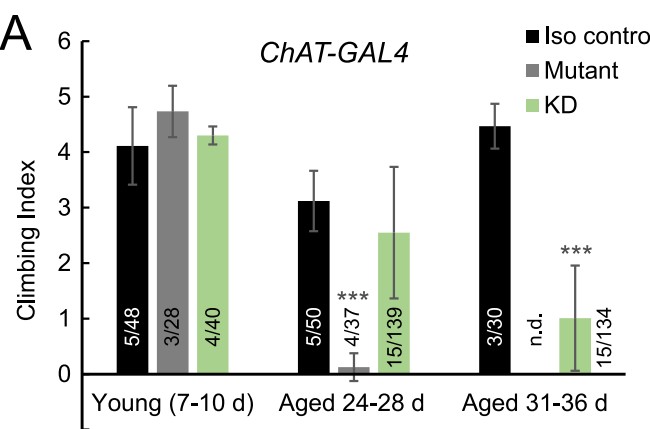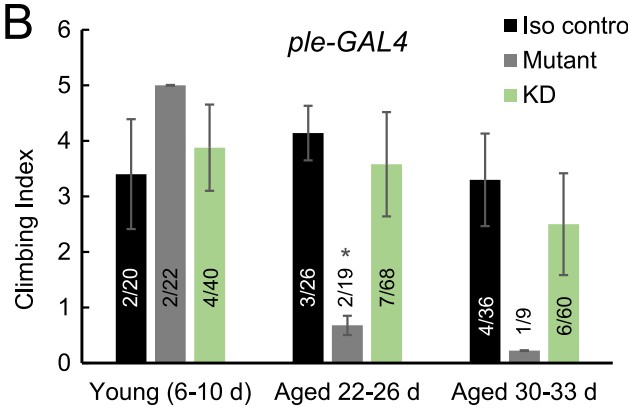

**Fig 1. Examples of student-generated data.** Climbing ability was tested in knockdown flies (green bars), as well as positive (*PLA2G6* null mutant, gray bars) and negative control flies (isogenic wild-type, black bars) at young age (≤ 10 days) and older age (> 20 days). *PLA2G6* mutant flies displayed severe climbing defects after 20 days of age. Knockdown in cholinergic neurons (A, *ChAT-GAL4*) but not dopaminergic neurons (B, *ple-GAL4*) resulted in reduced climbing in flies aged past 29 days. Averaged climbing indices are shown. Error bars are standard deviation. Lower numbers on the bars indicate the number of groups averaged; upper numbers indicate total number of flies assayed per condition. Statistical analysis by unpaired t-tests, as compared to negative controls. *$p < 0.01$, ***$p < 0.0001$.

and benefits gained (post-course only) that were representative of our three learning goals. We found that students reported large gains in benefits that represent thinking like scientists, such as "understanding how scientists work on real problems" (4.4) and "skill in the interpretation of results" (4.2) (Fig 3A). Related to communicating like scientists, students reported large gains in the skills of science writing and oral presentation, both as post-course benefits and in relation to pre-course experience (Fig 3B and 3D), with striking improvement compared to prior experience. Finally, students reported large gains in skills and qualities integral to performing like a scientist, such as learning laboratory techniques (4.3) and having tolerance for obstacles faced in the research process (4.2) (Fig 3C). They reported much to extensive gained experience in course elements such as reading primary literature (4.1), collecting and analyzing data (4.8 and 4.7, respectively), and participating in a project where students have input and the outcome is unknown (4.5 and 4.5) (Fig 3E), all of which are key aspects of authentic scientific research.

**Students do not report major clarification of career goals, but CURE participation results in broader benefits in career preparation.** Another aspect of the CURE survey asked students about their career aspirations pre- and post-course, because a possible outcome of CURE participation is recruitment to STEM careers [4, 12, 14, 15]. Most of our students (13 out of 14) reported their intent to pursue a career in the health professions at the beginning of the course (Fig 4A), consistent with the course being an upper-level elective for biology majors and with health professions being popular career goals at this institution. At the conclusion of the course, while some students did report changing their career goals (mostly within health professions categories), half the students reported that the course had not changed their career goals (Fig 4B). This is in accord with students rating an average small to moderate gain in "clarification of career path" as compared to how the students rated other benefits of the course (Fig 4C). However, there was one student who reported a change from "other graduate school to MS in science" and another student who changed from pursuing an MD to PhD degree. Despite no strong evidence that the CURE played a large role in clarifying career goals, our students still reported large gains in self-confidence and readiness for more demanding research, which we believe will benefit them as they continue to pursue their professional and academic training (Fig 4C).

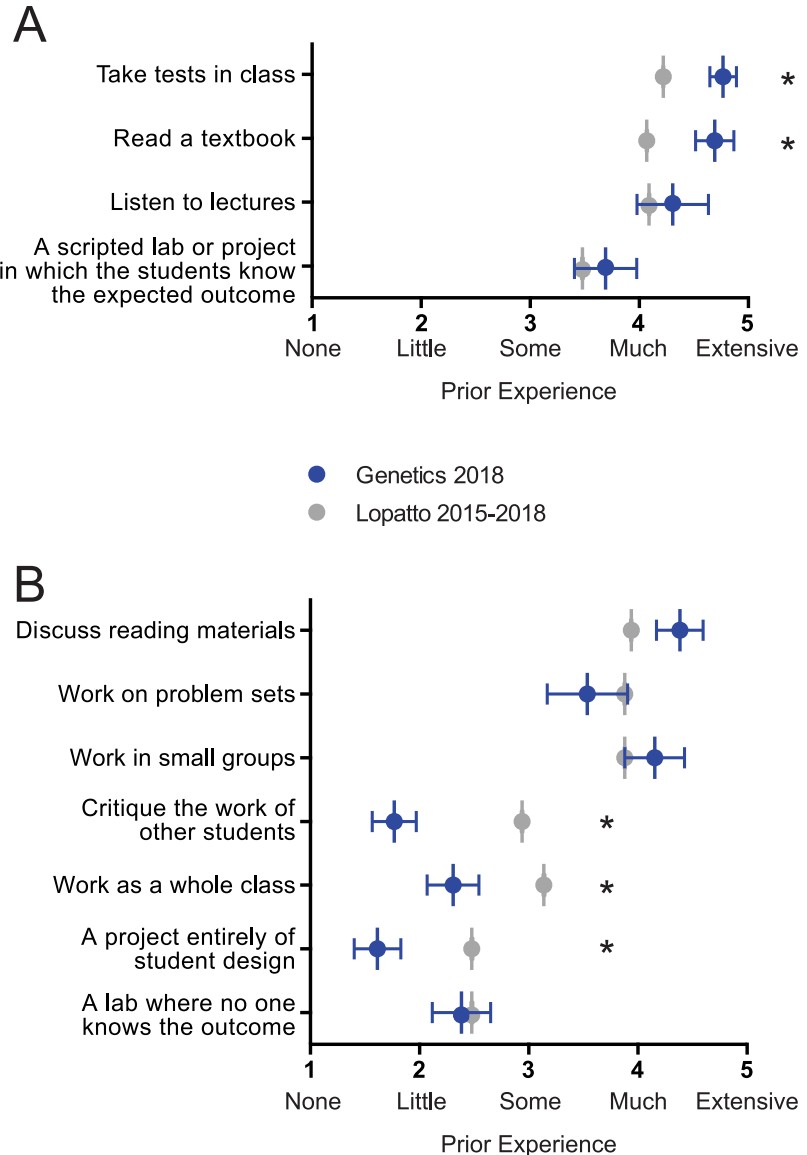

**Fig 2. Students report high levels of prior experience with traditional, non-active course elements on the CURE pre-course survey.** Students were asked to rate level of prior experience in 25 different course elements as none (1), little (2), some (3), much (4), extensive (5). Selected traditional, non-active course elements are shown in (A) and active course elements in (B). Mean response with standard error (n = 13) is shown in light blue, with the national mean from the CURE benchmark data shown in gray. One-sample t-tests were performed to compare national mean (as population mean) to our students' responses. *$p < 0.01$.

**Students display higher positive attitudes and lower negative attitudes towards science, both pre- and post-course.** Another possible effect of CURE participation is improved student attitudes towards science [13, 40]. In the CURE survey, students were asked to rate the degree to which they agree or disagree with various statements that reflect positive or negative attitudes towards science [13, 30]. We found that our students reported very high positive attitudes and low negative attitudes towards science (Fig 4D), especially in comparison to the national benchmark data (https://www.grinnell.edu/sites/default/files/docs/2019-07/CUREBenchmarkStatistics2015-2108.pdf). This was true pre- and post-course, suggesting that

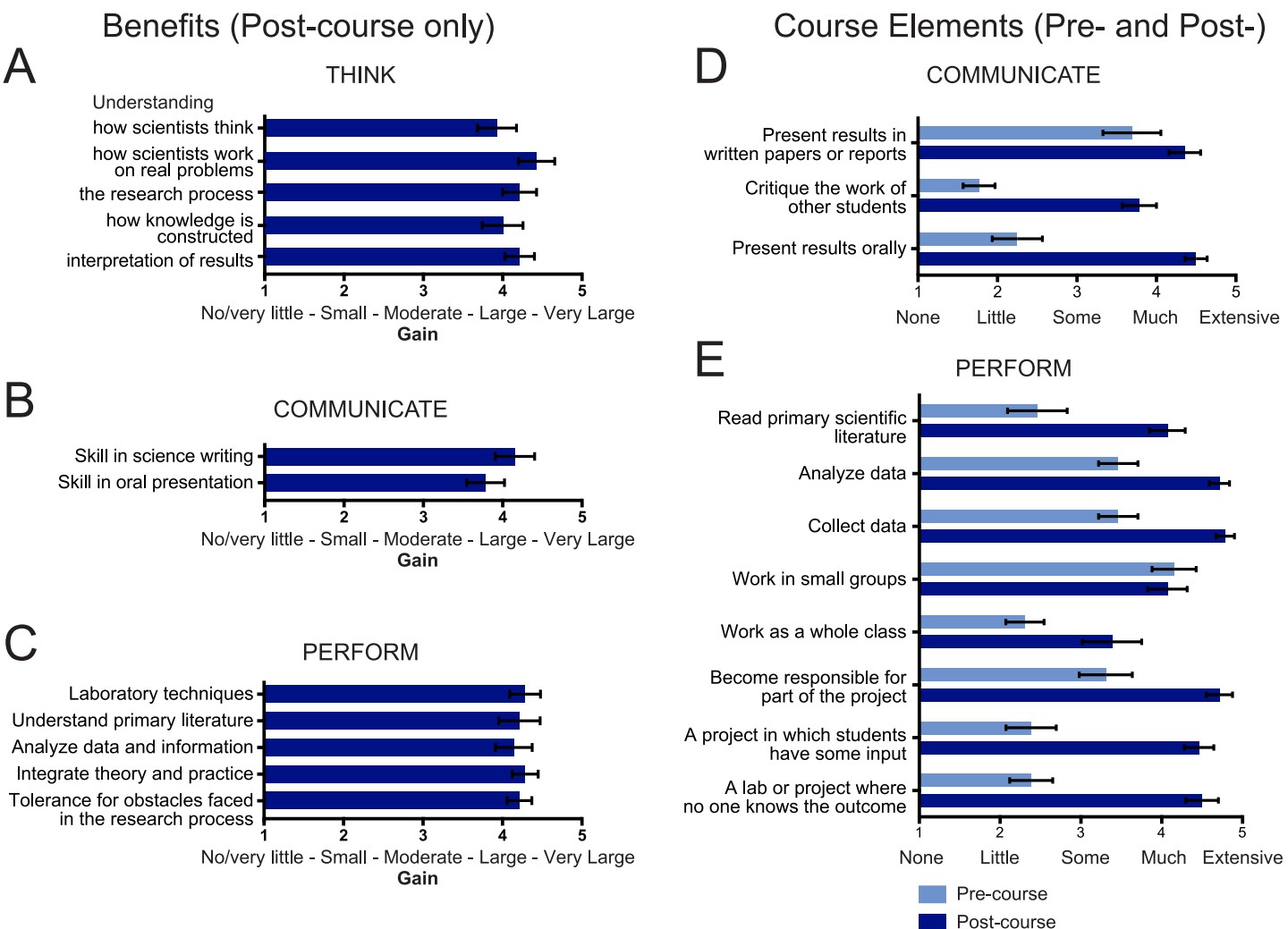

**Fig 3. Student CURE survey responses show high gains in learning goals of thinking, communicating, and performing like a scientist.** Students were asked to rate the amount of "benefits" gained in the post-course survey only as no/very little (1), small (2), moderate (3), large (4), and very large (5). Students were also asked to rate level of prior experience in 25 different "course elements" on the pre-course CURE survey and level of gained experience on the post-course survey as none (1), little (2), some (3), much (4), extensive (5). Selected benefits gained and course element experience/expertise relevant to learning how to "think" (A), "communicate" (B, D), and "perform" (C, E) like scientists are displayed (mean with standard error, n = 13 for course elements, n = 14 for benefits).

our students already had very positive existing attitudes towards science and that participation in the CURE did not markedly change them. There was a trend towards slightly lower negative attitudes post-course, but it was not significantly different from the pre-course average.

**Final lab report rubric analysis results.** In addition to examining students' self-reported gains through the CURE survey, we assessed their written lab reports for evidence of learning goal achievement. We developed a rubric (adapted from [33–37], see Supporting Information) and coded subsection elements into five different categories. The first three categories encompass components related to our three learning goals: think, communicate, and perform like a scientist. For example, each report's introduction was expected to provide context for the experiment, describe the model system, and justify and state the research question, elements that were coded as "scientific communication," as well as to summarize experimental goals and methods, coded as "scientific thinking." The last two rubric categories measured overall conceptual understanding of core biological principles and principles of the scientific method

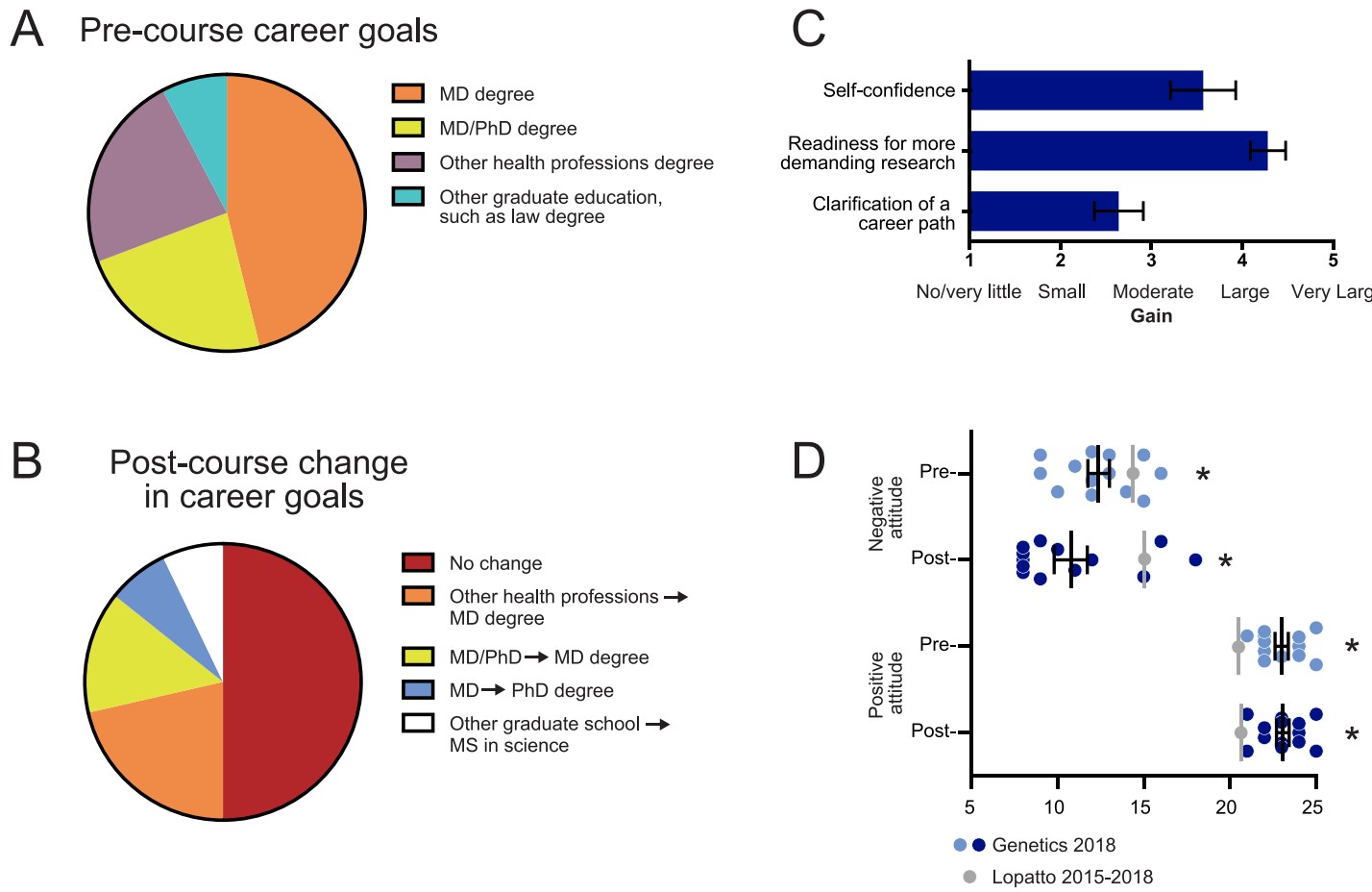

**Fig 4. Students report broader benefits to career goals on CURE survey.** Students were asked to report career goals pre-course (A, n = 13), and any change in career goals as a result of the CURE in the post-course survey (B, n = 14). Students were asked to rate the amount of "benefits" gained in the post-course survey only as no/very little (1), small (2), moderate (3), large (4), and very large (5). Selected benefits representative of broader benefits to career goals are shown in (C, mean, standard error, n = 14). (D) Positive and negative science attitudes were assessed (see Methods) in the pre- (light blue) and post- (blue) course survey (black bars represent mean ± standard error, n = 13). National benchmark mean is shown as a single data point (gray) and one-sample t-tests using national mean as population mean show statistical significance: *$p<0.01$. No significant differences were found between pre- and post-course science attitudes.

(adapted from the E-EDAT [36]). To simplify grading and analysis, each rubric component was graded with 0, 1, or 2 points. Examples of graded student responses are shown in Table 3. Overall, we found that students in the CURE displayed very high levels of achievement in the three learning goals (medians 83%, 90%, and 91% for "communicate," "perform," and "think" categories, respectively), with the highest scores in the "think" subsection (Fig 5A). Scores in the scientific method assessment were even higher, with a median of 96%. As a point of comparison, we also applied the rubric to 24 lab reports from the prior year, when students in the same course conducted a semester-long inquiry-based mapping project using *Drosophila* but were not engaged in authentic, open-ended research (Fig 5A, "2017 inquiry-based"). In both the 2017 and 2018 classes, the majority of students were upper-level biology majors on the pre-health track, suggesting that the two student populations were demographically similar (S1 Fig). Compared to the CURE, lab reports from the prior year scored significantly lower, with class medians of 52%, 55%, and 77% in the "communicate," "perform," and "think" categories, respectively (Fig 5A). Similarly, in the scientific method category, the 2017 class median was 77%, significantly lower than the CURE class median of 96%. Among the scientific method sub-elements (Fig 5B), students in the CURE showed significantly greater ability than students

**Table 3. Examples of graded rubric elements from lab reports.**

| Rubric element | Category coding | Example student response | Points awarded | Instructor comments |
|---|---|---|---|---|
| 6b. Reports all appropriate statistical analyses | Perform | *To determine how significantly different the results of the experimental climbing assays were from each control group, two two-tailed t-tests were performed assuming unequal variance. The results of the t-tests are displayed in Table 5.* | 2 | Student includes assumptions of the statistical test. |
| | | *Upon calculating the t-test it was concluded that the difference in climbing activity between the WT and Experimental flies was not significant (0.112). Subsequently, it was determined that, the difference in climbing activity among the WT and Mutant, as well as the Experimental and Mutant, was highly significant (t-test: 2.68042E-05 and 0.002042932 respectively). (See Table 4.)* | 1 | Student does not state the assumptions underlying the statistical test. |
| | | | 0 | Student does not include any statistical analysis |
| 7c. Provides multiple explanations for results, when appropriate. | Think | *There are several possibilities to explain these results. It can either be assumed that this data represents a false negative for a multitude of reasons, or alternatively, it can be deduced that glutamatergic neurons do not play a role in causing symptoms of PLAN. It is easy to assume that the results from this experiment were indicative of a lack of correlation between glutamatergic neurons and PLAN however in order to ensure that this is true, Drosophila can be bred to express iPLA2-beta-RNAi in all neurons except for glutamatergic neurons. If locomotor symptoms of PLAN develop in Drosophila under these conditions, then it can be assumed that glutamatergic neurons are not active in this disease pathology.* | 2 | Student explicitly states that there are multiple explanations for the observation that *PLA2G6* knockdown in glutamatergic neurons does not induce symptoms of neurodegeneration. |
| | | *The climbing indices for the experimental flies hinted at a tangible increase in climbing ability between 22–26 days and 32 days of age. This trend however, was not borne out quantitatively as shown via the T-test (see Results section)... While such is only speculatory, it could be suggested that octopaminergic and/or tyraminergic "fight-or-flight" neurons contribute to the temperance of the flies' inborn climbing proclivity and when those neuronal functionalities are removed, the flies are tempted to climb more. In order to bolster this assertion however, more research evaluating such a causal relationship would be necessary.* | 1 | Student observes that *PLA2G6* knockdown in octopaminergic and tyraminergic neurons led to slightly greater (but not statistically significant) climbing ability in aged flies compared to controls and suggests that this is due to those "fight-or-flight" neurons inhibiting climbing behavior when they are fully functional. He does suggest that this idea is "speculatory" and that "more research" is necessary but does not explicitly provide alternative explanation(s), e.g. that the knockdown did not affect climbing behavior, as indicated by the t-test. |
| | | *Given these data, it is reasonable to conclude that cholinergic neurons had a major contribution to the phenotypic climbing defect observed in the pan-neuronal iPLA2-beta knockout flies.* | 0 | Student does not provide multiple explanations for the observed result. |
| 7e. Includes a conclusion paragraph. | Communicate | *We showed that a decrease in production of iPLA2-beta protein in cholinergic neurons is linked to age dependent locomotion decline in flies. This study verified and added on to previous animal studies of iPLA2 (Shinzawa et al. 2008; Malik et al. 2008; Sumi-Akamaru et al. 2015; Iliadi et al. 2018; Steinhauer, unpublished data). In our results we did not see an exact relationship, in quantitative CI or onset of motor dysfunction, between our experimental knockdown line and the positive control null mutant line. While this may mean that there are other neurons which play a role in motor dysfunction caused by PLA2G6 KO, it is also possible that the results were due to inefficiency of RNAi system or that the Gal4-UAS system. What is certain from our results though, is that iPLA2-beta in cholinergic neurons play a role in locomotion, and that lack of the protein causes symptoms like that seen in PLAN patients. This will hopefully be helpful in future analysis of the mechanism behind PLA2G6 and its role in PLAN diseases.* | 2 | Final paragraph summarizes the experiment and results and provides broader context. |
| | | *There are a multitude of aspects that may have gone wrong, but we must keep trying for ourselves and the ones that came before us and for the ones who will come next.* | 1 | Final paragraph is generic and does not summarize the experiments or results. |
| | | *Also, our experiment focused on knocking out glutamatergic neurons. However, there are three types of VGLUTS in humans (Drosophila only have one of them): VGLUT1, VGLUT2, and VGLUT3. Each neuron can be found in different parts of the human brain; for instance, VGLUT1 is situated in the cerebellum (see diagram below). Therefore, we'd be interested in learning more about each of these subcategories of human glutamatergic neurons. We'd attempt to compare and contrast them to Drosophila VGLUT in order to fully comprehend how our Drosophila results relate to human VGLUT.* | 0 | Final paragraph raises interesting points for consideration when drawing conclusions about the experiment but does not summarize the experiment or results. |

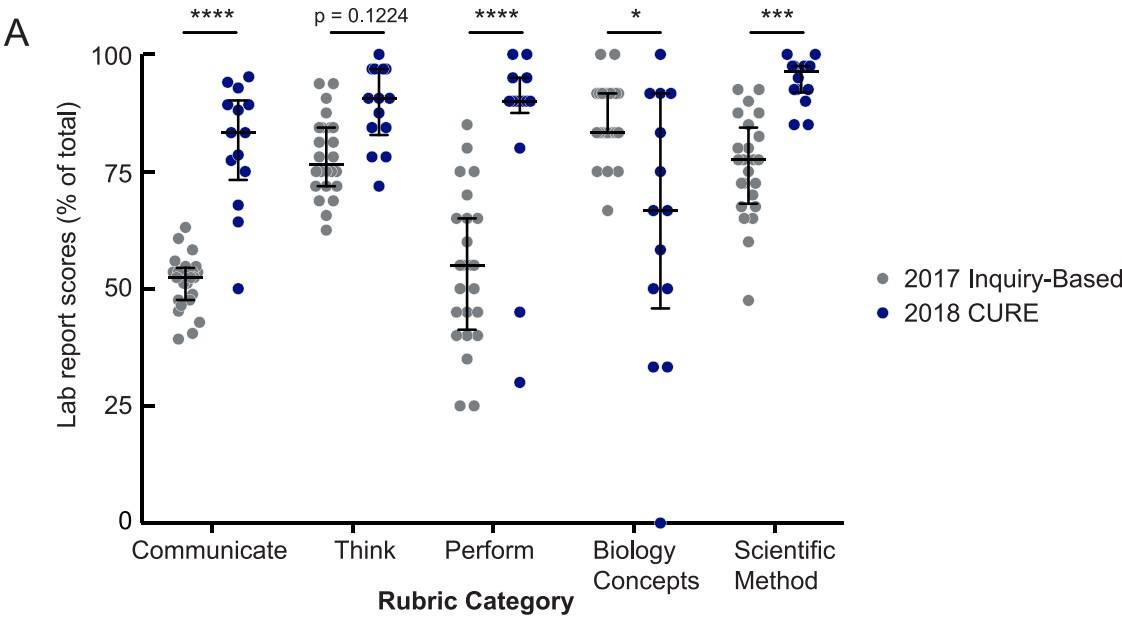

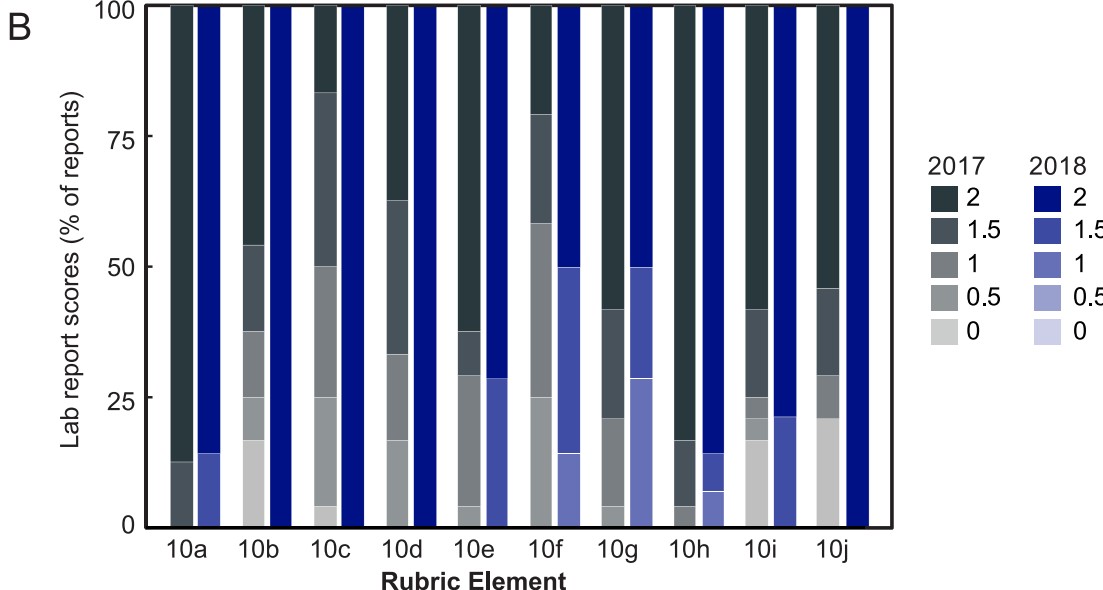

**Fig 5. Student written reports from the CURE lab display high achievement of learning goals.** (A) Written lab reports from the 2018 CURE lab (blue, n = 14) and prior year non-CURE lab ("2017 inquiry-based," gray, n = 24) were scored according to our rubric (see Methods) in five different areas: communicate, think, and perform like a scientist learning goals, conceptual understanding of biological principles relevant to the lab experiments, and conceptual understanding of scientific methods principles. Each score is reported as a percentage of the total possible points in that category. Bars are medians and interquartile ranges, statistical analysis by Kruskal-Wallis test with Dunn's correction for multiple comparisons. $^{****}p<0.0001$, $^{***}p<0.001$, $^{*}p<0.05$. (B) Lab reports from the 2018 CURE scored higher in rubric elements relevant to the scientific method (section 10). Blue bars represent percentage of 2018 lab reports that earned 0, 0.5, 1, 1.5, or 2 points on each rubric element (scores of 0.5 and 1.5 resulted from averaging the two independent graders' scores). Gray bars represent 2017 lab reports.

in the prior year in describing the biological rationale for their experiments (rubric element 10b), identifying independent and dependent variables (10c), describing how the dependent variable was measured (10d), identifying experimental and control groups (10f), and proposing future experiments (10j). Interestingly, both groups of students scored equally well in stating the experimental hypothesis (10a), understanding that other variables must be held constant (10e), understanding the importance of sample size (10g), and moderating their conclusions to account for multiple explanations or sources of error (10h). Finally, in contrast to the other rubric categories, scores from the 2018 CURE were significantly lower than those from the 2017 reports in the biological principles section (67% class median for 2018 compared to 83% class median for 2017, Fig 5A). This likely reflects the fact that the assessed biological concepts were different between the two years by necessity; the 2017 experiment centered on the concepts of genetic loci and recombination mapping, relatively simple concepts compared to the variety and complexity of concepts covered in the CURE (see Discussion).

**Narrative feedback.** We solicited anonymous written feedback from students at the conclusion of the course, and a few major themes emerged (Table 4). Many comments indicated preference for the CURE over traditional labs and implored the authors to offer more lab courses designed as CUREs, and no students said they preferred the traditional style of lab class. Many described feeling that their activities in traditional labs they had taken in the past were meaningless and inconsequential. In contrast, by participating in authentic, open-ended research through this CURE, they felt that the work was meaningful, and as a result, they were more engaged and motivated to learn. Many students said they valued being treated as "active participants," rather than "passive students" simply there for the grade, and they appreciated the level of responsibility they were given to carry out the experiments independently.

## Discussion

We conducted a genetics laboratory CURE to investigate neurodegeneration in *Drosophila* using climbing assays in conjunction with common, readily available genetic tools, including *GAL4-UAS* and RNAi. Experimental results may point to a prominent contribution of cholinergic neurons to the etiology of PLAN, which is somewhat unexpected given that motor decline in Parkinson's disease has been attributed primarily to the loss of dopaminergic neurons [41]. Indeed, recent experimental PLAN models have focused on dopaminergic neuron loss in mouse, zebrafish, and *Drosophila* [42–44]. Still, human PLAN patients display degeneration in multiple brain areas, and a role for cholinergic neurons in Parkinson's is beginning to be investigated [20, 41]. To understand the mechanism of motor decline in dystonia-parkinsonism and other PLAN diseases, future exploration of multiple neuronal subtypes is warranted, with possible applicability to other forms of parkinsonism as well. This CURE enriched the YU biology curriculum, in which traditional pedagogical approaches were over-represented and active inquiry-based approaches were lacking, and it involved students in a project with especially strong relevance to human disease and biomedical research. Students who took the CURE demonstrated large learning gains in thinking, communicating, and performing like scientists, as evidenced by both self-reporting in pre- and post-course CURE surveys and rubric-based analysis of written lab reports.

### Written lab reports demonstrate higher levels of scientific thinking, communication, and performance in the CURE than in the previous inquiry-based lab course

We developed a rubric, coded according to our learning goals of thinking, communicating, and performing like a scientist, to analyze students' final lab reports (Supporting Information).

**Table 4. Student narrative responses.**

| Category | Example Student Responses |
|---|---|
| Being an active participant | *The lab in general was a great experience for me. The lack of weekly quizzes, midterm, and final made it feel much more like I was an active participant in the lab as opposed a passive student there for the grade.* |
| | *There were multiple aspects of the lab which I enjoyed, but probably most of all was the general feel that I was not simply a student but an active participant in the lab.* |
| | *By doing one long original experiment I felt that I was being given actual responsibilities and being treated as a responsible adult.. . . The experiment lasted months and over that time I was able to grow invested in the research, and care about the accuracy of my results.* |
| Meaningfulness of contributing to original research | *It was both interesting and exciting to be able to participate in "real research" and have a final conclusion to present and discuss.* |
| | *Performing a novel, hands-on experiment in this lab taught me more about how research and the scientific method look than all of my other lab courses combined.* |
| | *I think its cool we were able to perfect our knowledge in one specific topic learning about drosophilla and feeling like our work made an actual impact.* |
| | *The fact that there were no pointless memorization quizzes or tests made me feel less like I was a student in a class and more like I was being trusted with actual research. Also, that we could set our own schedules, and go in and out of lab throughout the week, made me feel involved and invested in the lab work.* |
| | *Personally, I think there's nothing more exciting then contributing to neuroscience! And specifically research on neurodegenerative diseases. I was thrilled to take part in my professor's work. Even though my partner and I did not find results, I was happy that we found new evidence, this way the next hypothesis is more educated.* |
| Importance of primary literature review | *Also, I gained an immense amount from this class. The homework assignments in Lab, and the various other final assignments. . . gave me a valuable skill to be able to read and comprehend scientific papers in a timely manner which is something that I was terrible at before this semester.* |
| | *I thought the idea of having us read the papers relating to the techniques we performed was excellent and added a great dimension to the lab.* |
| Comparison to traditional style lab courses | *This lab was much better than other labs I have taken. Most lab courses are a series of demonstrations to show students ideas parallel to those taught in lecture. The demonstrations work to add a small dimension to lecture, but nothing about the scientific method or lab research is conveyed in those lab courses.* |
| | *In previous labs there was always a lab manual, with a different experiment every week, and we would show up to lab and blindly follow the experiment; few students actually understanding what was going on during the lab before writing the lab report. . . . I much preferred the style of the one long experiment as well as the freedom we were given with making our own schedule and access to the lab. . . . this system again, made me feel much more invested and involved in the lab, and not just passive. . . . This allowed me to actually like the lab, feel like I was actually accomplishing something interesting, and not feel like it was a waste of my time.* |
| | *This lab differed from other labs that I had taken before both in its emphasis on viewing primary literature and in the goal-oriented focus of the course.* |
| | *Each group performing different variations of the experiment added a sense of individuality to our experiments by further motivating and interesting us. Another pitfall of the typical lab course is that everyone performs the same experiment. None of the students feel invested in the lab as everyone is performing the same experiment and someone else will get good data, so why bother trying? The individual aspect of the experiment for each group gave us the feeling that we were each contributing a unique piece to the puzzle.* |
| | *This lab was better than previous labs because it was novel research and it felt more purposeful than just doing standard, rote experiments.* |
| | *I gained so much more from this lab than other labs. In other labs, you spend each week learning a different technique that you probably forget how to do after a few weeks.* |
| | *In previous labs, I had gone through the motions, showing up to lab having studied for a weekly quiz . . . would quickly do an experiment which I knew had been done countless times before, and get out of the lab as soon as possible. I never felt invested in any experiment I did and barely understood what was going on it. The general feel was that I was a student being tested on something as opposed to a researcher trying to figure something out. . . . Every week we had a new lab experiment to do, which did not necessarily relate to the previous or future labs, and this made each lab feel impersonal, pointless, and simply a waste of my time.* |
| | *. . .the general model which the genetics lab used, that of independence, less testing and memorization, and longer experiments could be implemented in creative ways in other undergraduate labs. On the memorization point, I think the goal of labs should be better defined. I believe that the goal of undergraduate lab courses should be to inspire students to get involved in scientific fields and encourage students to continue their pursuit of scientific knowledge and research.* |

*(Continued)*

**Table 4.** (Continued)

| Category | Example Student Responses |
|---|---|
| Critical feedback | *On the one hand I loved the hands off attitude and made sure I was on top of the flies and asked questions when I wasn't sure about something we needed to do or why we were doing something a certain way, some students may need more structure and guidance to ensure they do the proper things.* |
| | *I didn't enjoy getting negative results in our experiment. Although that is still an important goal, it was frustrating working for four or five months to only reach negative conclusion.* |
| | *Time commitment. I wish we could have gotten an extra credit for the course.* |

Two additional sections of the rubric assessed overall understanding of biological principles and the scientific method, the latter adapted from the E-EDAT [36]. Although several elements within the scientific method section were redundant with elements in other sections, we included it as a separate entity in order to assess students' general understanding of the key principles of experimental design, apart from the course particulars (see below, [36]). This rubric was developed ex post facto for analysis of our students' learning gains. However, it could be adapted easily for use as a summative assessment instrument in future iterations of this course. Indeed, we found it permitted precise and detailed evaluation of students' understanding, and its high inter-rater reliability (Cohen's weighted kappa = 0.71) demonstrates its utility in reducing grader bias.

Although the CURE lab reports demonstrated high achievement in most rubric categories, our analysis identified gaps in understanding the relevant biological principles (class median 67% in this category), which included the contribution of neuronal subtypes to neurodegenerative disease, the role of *PLA2G6* in neurodegeneration, the use of *Drosophila* as a model organism, the *GAL4-UAS* system (which incorporates transcriptional regulation and transgenic organisms), and RNA interference. We consider it a strength of the CURE that students were challenged to think about a bona fide modern experiment in context, but care must be taken to guide students' understanding to full fruition during class discussions. Formative assessments throughout the semester would furnish useful feedback to instructors and students before final lab reports are submitted.

Compared to the prior year's lab reports, which centered on an inquiry-based mapping procedure, the CURE lab reports demonstrated greater achievement in the three rubric categories of communicating, thinking, and performing like scientists, with median class scores above 83% in each category and cross-year differences in the communicate and perform categories reaching statistical significance. This comparison analysis suggests that the emphasis on bona fide experimentation and reporting in the CURE imparts strong benefits to students beyond those provided by other inquiry-driven exercises. However, because we cannot currently parse this key difference from other differences between the two years, including different experimental structures, different underlying principles, different students, different instructors (J.S. in year 1, R.D. in year 2), and a lack of primary literature incorporation in 2017, conclusions from our cross-year comparison may be limited. Notably, however, CURE students also demonstrated strikingly higher scores in the scientific method section of our rubric, with a class median of 96%, compared to 77% in 2017. This suggests that the CURE improved students' understanding of the scientific process and instilled key scientific skills that extend beyond the course particulars, including in describing the biological rationale for their experiments, in identifying independent and dependent variables as well as experimental and control groups, and in designing future experiments to distinguish between multiple explanations (Fig 5).

Thus, we believe it is reasonable to conclude that the CURE enhanced student learning above an inquiry-based project, particularly in areas related to the scientific process.

## CURE survey results and student narrative feedback suggest strong engagement with the project and direct learning benefits

We administered the widely used CURE survey developed by D. Lopatto to measure students' self-reported learning gains. Consistent with the lab report scoring, students reported large gains post-course in key aspects related to our learning goals of thinking, communicating and performing like a scientist (Fig 3). One possibility to consider is that our students were predisposed to have particularly positive outcomes in the CURE due to their higher positive and lower negative pre-course science attitudes than the national average (Fig 4). Although we cannot control for this sample bias, our student sample was at least free from volunteer bias because this was the first formal CURE at YU and the format was not advertised, thereby eliminating student self-selection for the CURE course format [45]. According to student narrative responses (see Table 4), being an active participant in authentic novel research with broader relevance was very personally meaningful and motivating for the students. Because students were responsible for generating a unique and novel dataset with the potential to contribute to a medical problem that carries acute emotional impact, feelings of ownership and accountability were high. This, in combination with the instructional emphasis on the scientific process, could explain the dramatic increase in learning gains [46]. The student narrative feedback provides a qualitative context for the positive results obtained from the CURE survey, which directly compares pre- and post-course responses, and the rubric-based lab report scoring, which does not rely on student self-reporting. All three forms of assessment taken together strongly support the conclusion that students benefited directly from the CURE.

CUREs and other inquiry-based exercises have been recommended as a mechanism to recruit and retain STEM students [2, 4]. Consistent with this proposal, our students reported large gains in readiness for more demanding research post-course (Fig 4C). However, we could not draw strong conclusions about the CURE's effectiveness in STEM recruitment and/ or retention because our student population consisted of mostly pre-health upper-level biology majors [15]. Our students were already committed to STEM careers, and thus 50% of the class reported no change in career goals following the course, and "clarification of a career path" showed only a very modest gain in the post-course benefits (Fig 4). This is consistent with mounting evidence that research experiences within the first two undergraduate years are more effective at promoting persistence in STEM fields than in later years [14, 15]. Still, we believe that our students' careers will benefit in other ways from participation in the CURE. In particular, the gains related to communicating like a scientist and interfacing with the primary literature will be critical as they pursue careers in medical fields that are rapidly advancing. As shown in Fig 4C, students also reported large gains in self-confidence, which can be a positive attribute in any career path. Future studies will need to follow larger cohorts of CURE students, including some from introductory level courses, as they progress through their undergraduate curriculum and matriculate into graduate programs to address the question of STEM recruitment and retention [4].

## Implementing the CURE

Understanding and combatting neurodegeneration is currently of paramount importance to medicine and human health, and animal models have proven to be extremely valuable in elucidating underlying genetic and molecular mechanisms [19, 47]. The climbing assay is a standard test of neurodegeneration in the *Drosophila* model system and can be performed by

novice students at minimal cost [28, 29]. Incorporating this experiment into a CURE provides students with the opportunity to participate in novel open-ended research with high impact potential and strong emotional appeal. We also designed our course to incorporate the key CURE elements of collaboration and iteration, with each student pair responsible for knocking down *PLA2G6* in only one subset of neurons, such that student-generated results were best understood in the context of the entire class dataset.

Students were required to attend to their flies on their own time and were given latitude to plan their experiments. This appeared to stimulate strong student engagement and provided the opportunity to perform like "real" scientists, but potentially at the expense of data consistency. While students in this CURE were able to generate interesting preliminary data that will guide future experiments, in order to generate higher fidelity data, stricter experimental guidelines may be necessary. In the narrative feedback responses, a few comments highlighted the time commitment and the high potential for negative results as negative attributes of the CURE (see Table 3). We recommend using class discussions to frame these aspects as part of the scientific process, while acknowledging the possibility that some students will discover that science is not their calling.

The design of this CURE experiment can be applied to other genes, environmental perturbations, or genetic or pharmacological screens, and it can be modified easily to accommodate larger class sizes (e.g., by using more *GAL4* lines, having multiple student pairs test overlapping genotypes, or having different student pairs test flies of different sexes or ages) and introductory students (e.g., by creating strict guidelines for experimentation). We have found that for those students inclined toward scientific inquiry, not only is the CURE an enriching and rewarding experience, but it also serves as a recruitment tool for apprentice-based research internships, with the added advantage that students are already trained in the techniques. Thus, this CURE, based on a simple physiological assay in *Drosophila*, has the potential to contribute novel findings to the field of neurodegeneration in a format that confers large student learning gains and tangential benefits to both students and faculty instructors.

## Supporting information

**S1 Fig. Student demographics in 2017 and 2018 are similar.** (A) A majority of students in both 2017 and 2018 classes were biology majors (blue). (B) A majority of students in both 2017 and 2018 classes were upper classmen (third year, orange, and fourth year, yellow).
(EPS)

**S1 File. Climbing assay protocol.** Adapted from [28, 29].
(DOCX)

**S2 File. Guided reading worksheet.** Adapted from Sarah Petersen, Kenyon College, personal communication to R.D.
(DOCX)

**S3 File. Figure/table analysis worksheet.** Adapted from [48].
(DOCX)

**S4 File. Journal club instructor guidelines.**
(DOCX)

**S5 File. Journal club discussion questions.** In reference to [49].
(DOCX)

**S6 File. Student guidelines for group presentations.**
(DOCX)

**S7 File. Instructor grading rubric for group presentations.**
(DOCX)

**S8 File. Student peer feedback form for group presentations.**
(DOCX)

**S9 File. Student guidelines for lab reports.** Adapted from [33].
(DOCX)

**S10 File. Instructor grading rubric for lab reports.** Adapted from [33–37].
(DOCX)

**S1 Dataset. CURE survey raw data.**
(XLSX)

**S2 Dataset. Lab report scoring raw data.**
(XLSX)

## Acknowledgments

We thank Jennifer Kennell, Elizabeth Genné-Bacon, Erika Crispo, and David Lopatto for critical comments on the manuscript; the Albert Einstein College of Medicine IRB for oversight; Liam Eliach and Raymond Reynoso for assistance in the lab; and Mimi Shirasu-Hiza for mentorship. Stocks obtained from the Bloomington *Drosophila* Stock Center (NIH P40OD018537) were used in this study.

## Author Contributions

**Conceptualization:** Rebecca Delventhal, Josefa Steinhauer.

**Data curation:** Rebecca Delventhal, Josefa Steinhauer.

**Formal analysis:** Rebecca Delventhal, Josefa Steinhauer.

**Funding acquisition:** Josefa Steinhauer.

**Investigation:** Rebecca Delventhal, Josefa Steinhauer.

**Methodology:** Rebecca Delventhal, Josefa Steinhauer.

**Project administration:** Rebecca Delventhal, Josefa Steinhauer.

**Resources:** Rebecca Delventhal, Josefa Steinhauer.

**Supervision:** Josefa Steinhauer.

**Writing – original draft:** Rebecca Delventhal, Josefa Steinhauer.

**Writing – review & editing:** Rebecca Delventhal, Josefa Steinhauer.

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
