## [Decision Letter · Decision Letter 0]

14 Jan 2020

PONE-D-19-34172

A Course-based Undergraduate Research Experience examining neurodegeneration in Drosophila melanogaster teaches students to think, communicate, and perform like scientists

PLOS ONE

Dear Dr. Steinhauer,

Thank you for submitting your manuscript to PLOS ONE. After careful consideration, we feel that it has merit but does not fully meet PLOS ONE’s publication criteria as it currently stands. Therefore, we invite you to submit a revised version of the manuscript that addresses the points raised during the review process.

Overall, both reviewers felt that the CURE course analyzed in your manuscript had strong supporting data and indications of success, indicating that it should be published and the data shared.   Nevertheless, both reviewer also made several constructive comments that I believe will greatly enhance the value of of this paper.   There are not many of these comments and most are minor, but I would like all of them to be address before acceptance.   CURE courses are a timely topic, and I am pleased that you are seeking  publication of your analysis of the course outcomes in an open access research journal.  

We would appreciate receiving your revised manuscript by Feb 28 2020 11:59PM. To enhance the reproducibility of your results, we recommend that if applicable you deposit your laboratory protocols in protocols.io, where a protocol can be assigned its own identifier (DOI) such that it can be cited independently in the future. For instructions see: http://journals.plos.org/plosone/s/submission-guidelines#loc-laboratory-protocols

We look forward to receiving your revised manuscript.

Kind regards,

Gregg Roman, PhD

Academic Editor

PLOS ONE

Journal Requirements:

Reviewers' comments:

Reviewer's Responses to Questions

**Comments to the Author**

1. Is the manuscript technically sound, and do the data support the conclusions?

Reviewer #1: Partly

Reviewer #2: Yes

2. Has the statistical analysis been performed appropriately and rigorously? 

Reviewer #1: Yes

Reviewer #2: Yes

3. Have the authors made all data underlying the findings in their manuscript fully available?

Reviewer #1: No

Reviewer #2: Yes

4. Is the manuscript presented in an intelligible fashion and written in standard English?

Reviewer #1: Yes

Reviewer #2: Yes

5. Review Comments to the Author

Reviewer #1: Overall this is an interesting CURE research topic with a substantial amount of evidence that it was successful in achieving the authors learning goals. The authors outline an engaging and relevant CURE research topic that could be used in numerous settings. However there are several areas where I feel that this manuscript could be enhanced, I have outlined my specific comments below.

Line 48 The authors list their learning goals as innovative, but there is little “innovative” about helping students learn to think, communicate and/or perform like scientists.

Line 124. “filled a previously unaddressed niche….”. The authors provide little data to support any conclusions about the overall structure of the curriculum, nor do they suggest exactly what niche they are trying to fill.

Lines 248 and 260 in both locations the authors mention the number of flies used in the trials, but describe this differently (6-12 or ~10). Consistency here would be helpful

Line 258 the authors state that 15 students started the course, but only 14 completed it. This leaves open the question of why the one student didn’t complete the course. Was the reason for this relevant to this study?

Lines263, 266, and 268. My preference would be to write out “5 out of 7” as opposed to “5/7”, etc.

Lines 349-376. Student clarification of career goals. This section focuses on a topic which is not part of the course learning goals nor discussed in the introduction. I know that is is part of the survey, but I am not sure why the authors have included it. I would delete it from this manuscript.

Line 391-442. Analysis of written reports. The analysis of these data is fine, but without knowing more about the make-up of the students in 2018 vs 2017 it is very hard to know how to interpret the results. The 2017 group was almost twice as large (24 vs 14) and no data are presented about their academic backgrounds, majors, intended careers, etc are provided to allow the reader to evaluate whether these two groups of students are similar or not.

Lines 469-472 See comment line 124 above.

Line 513-514. The authors suggest that this CURE “successfully” trained students in the scientific process. Given that this assessment is based the methods section of a lab report, I think increased their training would be more appropriate. Successful training in the scientific process is more than just methods.

I was unsuccessful in accessing the data based on the Qualtrics URL that was provided.

Reviewer #2: This is a well written manuscript which adds to the growing acceptance of CUREs as a tool to engage and train students in the scientific process using active methodologies. The manuscript utilizes simple genetic assays to identify neuronal subtypes where down regulation of a particular gene, PLA2G6 results in neuronal defects. I believe the paper as presented is acceptable for publication in PLOS One but only after some minor points have been taken care of.

1) Line 160-161: Why were the students not informed that they would be conducting an authentic research prior to the first day of class? How is this advantageous?

2) Line 237: Please use RRIDs when describing the stocks used. For more details please see: https://scicrunch.org/resources

3) Line 248: Why was 20 seconds chosen for climbing assays? Provide reasons or cite a reference if one is available.

4) Line 261: >20 days? Is there a limit? 20-29 days as later used? Perhaps specifying this here makes it clearer?

5) Figure quality was not the best when the submitted PDF was printed. Please make sure that the figures are clear and of high quality.

6) Some of the figures used colors that are very close to each other on the spectrum making it difficult to distinguish. For example: Figure 4A all shades of green. Similarly figure 4B has colors which are very close to each other on the spectrum.

6. PLOS authors have the option to publish the peer review history of their article (what does this mean?). If published, this will include your full peer review and any attached files.

Reviewer #1: No

Reviewer #2: No

---

## [Author Response · Author response to Decision Letter 0]

18 Feb 2020

February 17, 2020

Gregg Roman, PhD

Academic Editor

PLoS One

Dear Dr. Roman,

We thank the reviewers for their time, valuable comments, and suggestions. We are grateful that the reviewers found our work on a Drosophila-based CURE laboratory to be of interest and substantiated by the evidence presented. We appreciate the suggested improvements and have responded below to every concern mentioned. We hope you agree that the manuscript has been strengthened and clarified by the changes. 

 Sincerely,

 Josefa Steinhauer, Ph.D.

 Associate Professor

 Department of Biology

 Yeshiva College

 

Response to reviewer comments

Reviewer #1: 

• Line 48 The authors list their learning goals as innovative, but there is little “innovative” about helping students learn to think, communicate and/or perform like scientists.

We have removed the descriptor of our learning goals as “innovative.”

• Line 124. “filled a previously unaddressed niche….”. The authors provide little data to support any conclusions about the overall structure of the curriculum, nor do they suggest exactly what niche they are trying to fill.

We clarified the language as follows:

Line 136: “Overall, we found that the CURE generated large student learning gains, fostered active learning, which is lacking within our biology major curriculum, and improved student performance in written assignments above the prior inquiry-driven lab course.”

Additionally, we have highlighted the evidence presented in Fig. 2 that students had little experience with active learning in their prior curriculum as follows: 

Lines 349-351: “Because most of the students in this course were upper-level biology majors (S1 Fig), these results suggest that the YU biology curriculum was lacking active learning components prior to the implementation of this CURE.”

• Lines 248 and 260 in both locations the authors mention the number of flies used in the trials, but describe this differently (6-12 or ~10). Consistency here would be helpful

To clarify, students were instructed to collect about 10 flies in each group, and the actual numbers used ranged from 6-12. We have amended the language at former line 260 (now line 285) as follows:

“Students collected and assayed at least four groups of 6-12 flies each.”

• Line 258 the authors state that 15 students started the course, but only 14 completed it. This leaves open the question of why the one student didn’t complete the course. Was the reason for this relevant to this study?

One student dropped the course late in the semester (he completed all experimental aspects of the lab) for reasons unrelated to the CURE- he was concerned about his grade on the midterm exam in the lecture portion of the course (taught by a different instructor) and was pre-health bound. Because these reasons were unrelated to the CURE, we have simplified the language in the manuscript as follows on line 284: 

“Fourteen students completed the course, and they worked in self-selected pairs for the entirety of the semester.”

• Lines 263, 266, and 268. My preference would be to write out “5 out of 7” as opposed to “5/7”, etc.

We have corrected this language as requested.

• Lines 349-376. Student clarification of career goals. This section focuses on a topic which is not part of the course learning goals nor discussed in the introduction. I know that is is part of the survey, but I am not sure why the authors have included it. I would delete it from this manuscript.

We appreciate this comment for highlighting an area of the manuscript that required more explanation, which we have expanded and clarified in the Introduction (line 77) and Discussion (lines 623-640). As we state in the opening sentence of the Introduction (line 67-70), one of the most common and prominent rationales for CUREs is to promote student persistence in STEM majors and careers. Thus, we believe it is important to include these data and discussion because they help place our work in the context of the CURE literature. Although we did not observe an effect of our CURE on STEM recruitment or retention, this result is consistent with published observations suggesting that research experiences within the first two years of college have a larger effect than upper level experiences. We hope that our revised text and additional references clarify and strengthen this point.

• Line 391-442. Analysis of written reports. The analysis of these data is fine, but without knowing more about the make-up of the students in 2018 vs 2017 it is very hard to know how to interpret the results. The 2017 group was almost twice as large (24 vs 14) and no data are presented about their academic backgrounds, majors, intended careers, etc are provided to allow the reader to evaluate whether these two groups of students are similar or not.

We appreciate the reviewer’s concern about potential demographic differences between students in the 2017 and 2018 classes, and we have included data on their levels and majors in a new Figure S1 (lines 361-363). Twenty-three out of 24 of the 2017 students were Biology majors, and all 14 2018 students were Biology or Biochemistry majors. Both years included students in their 2nd, 3rd, and 4th years, and although fewer students in the 2017 class were in their 2nd year compared to 2018, we do not expect this small difference to impact performance on the lab reports (if anything, we might have expected the younger students to have lower scores on the lab reports, but the 2018 students scored higher across the board). While we don’t have precise data on intended careers for the 2017 students, as they did not take the CURE survey, we consulted with the Yeshiva College Pre-Health Advising Office and confirmed that 23 out of 24 students were pre-health bound, very similar to the students who took the course in 2018.

• Lines 469-472 See comment line 124 above.

Similarly, we have amended the language as follows on lines 540-541: “This CURE enriched the YU biology curriculum,...”

• Line 513-514. The authors suggest that this CURE “successfully” trained students in the scientific process. Given that this assessment is based the methods section of a lab report, I think increased their training would be more appropriate. Successful training in the scientific process is more than just methods.

We appreciate the suggestion and have amended the wording as follows:

Lines 587-588: “This suggests that the CURE improved students’ understanding of the scientific process.” 

We also have added text clarifying that the “scientific method” section of the rubric was coded to statements throughout the entirety of the lab reports beyond the lab reports’ methods sections (lines 242-244).

• I was unsuccessful in accessing the data based on the Qualtrics URL that was provided.

We apologize for the fact that the Qualtrics data are password protected and have now included the CURE survey raw data, as well as the lab report scoring data, as Excel spreadsheets in the supporting information. 

Reviewer #2: 

1) Line 160-161: Why were the students not informed that they would be conducting an authentic research prior to the first day of class? How is this advantageous?

The students were not informed of the CURE lab component of the Genetics course prior to the beginning of the semester, which may reduce volunteer bias for the format. There is a concern in the CURE literature (for example, see Brownell at al. JMBE 2013) that students who are more inclined to research might opt to take a CURE course, whereas students who don’t think they’re interested or capable will decline to enroll. Our student population did not have the opportunity to self-select, hopefully minimizing the effect that could have on our student outcomes. 

This point is further clarified in the text at lines 609-612: “Although we cannot control for this sample bias, our student sample was at least free from volunteer bias because this was the first formal CURE at YU and the format was not advertised, thereby eliminating student self-selection for the CURE course format (44).”

2) Line 237: Please use RRIDs when describing the stocks used. For more details please see: https://scicrunch.org/resources

We have updated the methods to include RRIDs in addition to the BDSC number for fly stocks.

3) Line 248: Why was 20 seconds chosen for climbing assays? Provide reasons or cite a reference if one is available.

Twenty seconds was empirically determined, and a citation from the author (JS) using this time interval was added. While shorter climbing assay durations are sometimes reported in the literature, aged flies typically need longer to climb due to reduced locomotor ability even in wild-type flies. 

4) Line 261: >20 days? Is there a limit? 20-29 days as later used? Perhaps specifying this here makes it clearer?

We have clarified that students were instructed to test flies at 22-28 days old.

5) Figure quality was not the best when the submitted PDF was printed. Please make sure that the figures are clear and of high quality.

We have double-checked the original images to ensure that they are the correct resolution.

6) Some of the figures used colors that are very close to each other on the spectrum making it difficult to distinguish. For example: Figure 4A all shades of green. Similarly figure 4B has colors which are very close to each other on the spectrum.

We appreciate the reviewer’s comment and have updated the colors throughout the figures to enhance contrast and clarity. In particular, we have made the pie charts in Fig 4 multicolored to ease distinction.

---

## [Decision Letter · Decision Letter 1]

12 Mar 2020

A Course-based Undergraduate Research Experience examining neurodegeneration in Drosophila melanogaster teaches students to think, communicate, and perform like scientists

PONE-D-19-34172R1

Dear Dr. Steinhauer,

We are pleased to inform you that your manuscript has been judged scientifically suitable for publication and will be formally accepted for publication once it complies with all outstanding technical requirements.

With kind regards,

Gregg Roman, PhD

Academic Editor

PLOS ONE

Additional Editor Comments (optional):

Reviewers' comments:

Reviewer's Responses to Questions

**Comments to the Author**

1. If the authors have adequately addressed your comments raised in a previous round of review and you feel that this manuscript is now acceptable for publication, you may indicate that here to bypass the “Comments to the Author” section, enter your conflict of interest statement in the “Confidential to Editor” section, and submit your "Accept" recommendation.

Reviewer #1: All comments have been addressed

Reviewer #2: All comments have been addressed

2. Is the manuscript technically sound, and do the data support the conclusions?

Reviewer #1: Yes

Reviewer #2: Yes

3. Has the statistical analysis been performed appropriately and rigorously? 

Reviewer #1: Yes

Reviewer #2: Yes

4. Have the authors made all data underlying the findings in their manuscript fully available?

Reviewer #1: Yes

Reviewer #2: Yes

5. Is the manuscript presented in an intelligible fashion and written in standard English?

Reviewer #1: Yes

Reviewer #2: Yes

6. Review Comments to the Author

Reviewer #1: The changes that the authors made have both addressed the reviewer comments/concerns and have strengthened the manuscript.

Reviewer #2: I had difficulty finding lines 609-612 (as mentioned in the response letter) but found where the change had been made at Line 543. This is acceptable to me. The paper in my view is ready for acceptance.

7. PLOS authors have the option to publish the peer review history of their article (what does this mean?). If published, this will include your full peer review and any attached files.

Reviewer #1: No

Reviewer #2: No

---

## [Editor Report · Acceptance letter]

16 Mar 2020

PONE-D-19-34172R1 

A Course-based Undergraduate Research Experience examining neurodegeneration in *Drosophila melanogaster* teaches students to think, communicate, and perform like scientists 

Dear Dr. Steinhauer:

I am pleased to inform you that your manuscript has been deemed suitable for publication in PLOS ONE. Congratulations! Your manuscript is now with our production department. 

With kind regards,

on behalf of

Dr Gregg Roman 

Academic Editor

PLOS ONE